# Development of a Novel In Silico Classification Model to Assess Reactive Metabolite Formation in the Cysteine Trapping Assay and Investigation of Important Substructures

**DOI:** 10.3390/biom14050535

**Published:** 2024-04-30

**Authors:** Yuki Umemori, Koichi Handa, Saki Yoshimura, Michiharu Kageyama, Takeshi Iijima

**Affiliations:** DMPK Research Department, Teijin Institute for Bio-Medical Research, TEIJIN PHARMA LIMITED, 4-3-2 Asahigaoka, Hino-shi, Tokyo 191-8512, Japan; yuki.umemori@axcelead-twp.com (Y.U.); saki.yoshimura@axcelead-twp.com (S.Y.); michiharu.kageyama@axcelead-twp.com (M.K.); t.iijima@teijin.co.jp (T.I.)

**Keywords:** cysteine trapping assay, drug-induced liver injury (DILI), idiosyncratic DILI, hepatotoxicity, message passing neural network, random forest, substructures, QSAR, toxicity, reactive metabolite

## Abstract

Predicting whether a compound can cause drug-induced liver injury (DILI) is difficult due to the complexity of drug mechanism. The cysteine trapping assay is a method for detecting reactive metabolites that bind to microsomes covalently. However, it is cumbersome to use 35S isotope-labeled cysteine for this assay. Therefore, we constructed an in silico classification model for predicting a positive/negative outcome in the cysteine trapping assay. We collected 475 compounds (436 in-house compounds and 39 publicly available drugs) based on experimental data performed in this study, and the composition of the results showed 248 positives and 227 negatives. Using a Message Passing Neural Network (MPNN) and Random Forest (RF) with extended connectivity fingerprint (ECFP) 4, we built machine learning models to predict the covalent binding risk of compounds. In the time-split dataset, AUC-ROC of MPNN and RF were 0.625 and 0.559 in the hold-out test, restrictively. This result suggests that the MPNN model has a higher predictivity than RF in the time-split dataset. Hence, we conclude that the in silico MPNN classification model for the cysteine trapping assay has a better predictive power. Furthermore, most of the substructures that contributed positively to the cysteine trapping assay were consistent with previous results.

## 1. Introduction

The success rate of new drug development is low, and one of the major causes of failure in clinical trials is toxicity [1]. Hepatotoxicity or cardiotoxicity accounts for most of these failures [2]. Since clinical drug development is very expensive, it is important for pharmaceutical companies to predict the toxicity of candidate compounds in the early stages of drug discovery to exclude compounds with risks.

Hepatotoxicity, namely drug-induced liver injury (DILI), in clinical trials includes intrinsic DILI and idiosyncratic DILI. Intrinsic DILI is dose-dependent and can often be avoided with dose adjustment [3]. In contrast, idiosyncratic DILI is dose-independent and depends on the patient’s genetic background, so its incidence is rare [4,5,6], and thus it is extremely difficult to predict its occurrence. In some instances, idiosyncratic DILI is discovered after a product is already on the market, leading to drug withdrawal [7]. Recently, there was a case of fasiglifam-induced liver damage in the late clinical stage, leading to discontinuation of the clinical trial [8].

Based on the mechanism, idiosyncratic DILI is classified into hepatocellular, cholestasis, and mixed types. Although the detailed molecular mechanisms of these are still largely unknown, in hepatocellular injury, reactive metabolites covalently bind to proteins, which are then processed by antigen-presenting cells, leading to abnormal compound-specific T-cells. The derived “hapten concept” is well known [9]. In the cholestatic form, drugs can inhibit the bile salt export pump (BSEP), leading to the accumulation of bile acids in hepatocytes, resulting in toxicity [10,11]. In addition, the generation of reactive oxygen species (ROS) and mitochondrial disorders are intricately intertwined [12]. By screening compounds in the early stages of drug discovery based on these mechanisms that have been recently clarified, we can avoid fatal outcomes in the late stages of marketing and clinical trials.

To experimentally evaluate idiosyncratic DILI during the nonclinical stage, a method was developed in the 2000s to measure the amount of covalent binding to microsomal or hepatocyte proteins in vitro using radiolabeled compounds. This method uses the amount of covalent binding as a quantitative index of reactive metabolites, and also examines the relationship with the clinical dose [13,14,15]. In addition, it was also shown that quantification was improved by incorporating multifaceted indicators, such as transporter inhibitory activity of proteins like BSEP and mitochondrial damage [16]. An assay for quantitatively estimating the covalent bonding ability of this compound to a protein requires the use of a radiolabeled compound, which is difficult to handle in the initial stages of drug discovery due to costs incurred and convenience. Glutathione (GSH) systems using fluorescence have also been proposed [17], but at present, tests using radiolabeled substrates (such as [^35^S] cysteine or [^14^C] cyanide) are conducted due to their high sensitivity [18,19]. By using such experimental systems, it is possible to evaluate idiosyncratic DILI in the early stages of drug discovery, although it is limited to the hepatocellular disorder type. For the cholestasis type, an experimental system using a sandwich culture method that forms bile ducts in vitro has been established [20], although it has not yet attained a predictive power suitable for screening in the early stages of drug discovery due to the difficulty of the experiment and costs. Hence, to improve our ability to forecast the possibility of various forms of hepatotoxicity at an early stage, prediction by computational methods would be ideal.

A consortium of pharmaceutical companies is currently developing DILI-Sim^®^, which is a computational prediction method for DILI [21]. However, although DILI-Sim is excellent for prediction in the late clinical stage and for already-marketed products [overdosed acetaminophen [22], ubrogepant comparing with telecagepant and MK-3207 [23], a sublingual formulation of riluzole [24], tolvaptan and its metabolite [25], the amount of input information required is too large to apply it at the drug discovery stage. In contrast, in silico models using various machine learning algorithms have been developed for DILI prediction in the early stages of drug discovery. Until now, the development of prediction methods based on classical methods using the structure of compounds has been vigorously pursued [26,27,28,29]; however, there are also many multimodal methods being developed that combine various explanatory variables such as calculated values (target protein prediction) [30] or experimental values (gene expression, protein expression, imaging data, etc.) [29,31,32]. It should be noted that the dataset used (called LTKB) was labeled with clinical information (casualty, incidence, severity from trials, literature surveys, and reports [33]. Considering the difficulty of predicting DILI in clinics [34], intrinsic DILI and idiosyncratic DILI might overlap, making the task of predicting DILI much more difficult. Furthermore, recently, various machine learning approaches have been proposed for searching for optimal models by combining multiple algorithms and multiple descriptors [35], converting complex toxicity information into integrated descriptors [36], and precise prediction by multi-binary classification models [37].

In this study, we decided not to create a direct prediction model for DILI, considering that the mechanism of DILI is diverse. Instead, we focused on predicting the formation of reactive metabolites that can predict idiosyncratic DILI with high probability. It is possible to find a machine learning model for predicting the site of GSH conjugates [38]. Here, to achieve quantitative prediction, specifically, using the ^35^S-cysteine trapping data acquired in-house, we created an in silico binary classification model using machine learning. Although it is not possible to directly predict DILI, this model can predict the presence or absence of reactive metabolites, which are a crucial predictor of idiosyncratic DILI. In addition, to make a more practical contribution to drug discovery, we also estimated the substructures that cause the generation of reactive metabolites.

## 2. Materials and Methods

### 2.1. Experimental Method

#### 2.1.1. Sample Preparation

The reagents for making the reaction solution were phosphate buffer (pH 7.4; 100 mM), ethylenediaminetetraacetic acid (EDTA; 1 mM) (Tokyo Kasei, Tokyo, Japan), MgCl_2_ hexahydrate (Fuji Film Wako Co., Ltd., Osaka, Japan, reagent special grade; 3 mM), glucose-6-phosphate (G6P; 5 mM) (Roche, Basel, Switzerland), G6P dehydrogenase (G6PDH; 1 IU) (Oriental Yeast Co., Ltd., Tokyo, Japan), human liver microsome solution (Xenotech, Kansas City, MO, USA; 2 mg/mL), and L-[35S]-Cysteine (PerkinElmer Japan Co., Ltd., Kanagawa, Japan; 3.7 MBq/mL), which were prepared using ultrapure water. The evaluation compound was dissolved in dimethyl sulfoxide (DMSO) and its concentration was adjusted to 4 mM. This was used as a substrate solution.

#### 2.1.2. Reaction in Microsome

On ice, 239 µL of the reaction solution was dispensed into a glass tube and then 1.25 µL of the substrate solution was added, mixed, and placed in a water bath (PERSONAL-11, Taitec Co., Ltd., Saitama, Japan) at 37 °C. After pre-incubation for 5 min, 10 µL of 25 mM nicotinamide adenine dinucleotide phosphate (NADPH) (Roche, Basel, Switzerland) solution was added and mixed to a final concentration of 20 µM to initiate a metabolic reaction and incubated in a water bath at 37 °C for 2 h. After incubation, 500 µL of acetonitrile was added and mixed, and the tube was chilled on ice to deproteinize, after which 50 µL of 250 mM GSH solution was added and mixed to stop the reaction, and centrifuged (approximately 1500× *g*, 4°C, 10 min, CF16RXII, Hitachi, Ltd., Tokyo, Japan). After centrifugation, 750 µL of the supernatant was transferred to a new glass tube and concentrated by vacufugation (VC-96N, Taitec Co., Ltd., Saitama, Japan) for 2 h. After concentrating the solution, we added 100 µL of the redissolving solution (40% methanol solution), mixed it with a vortex (IS-MBI minimixer, Ikeda Rika Co., Ltd., Tokyo, Japan) to redissolve, and then the sample was placed in a filter tube [Merck Millipore, Darmstadt, Germany] and centrifuged (approximately 5000× *g*, room temperature, 10 min).

#### 2.1.3. Measurement by HPLC with Radiomatic Detector

Both High Performance Liquid Chromatography (HPLC) (Nexera XR, Shimadzu Corporation, Kyoto, Japan) and a radiomatic detector (Radiomatic 610TR, PerkinElmer Japan Co., Ltd.) were used to analyze the prepared samples. The 35S content was detected by running the separated sample via the 45-min analytical method through a radiomatic detector. The mobile phase was composed of a gradient of 0.1% (*v*/*v*) formic acid/water and 0.1% (*v*/*v*) formic acid/acetonitrile, and the chromatographic separation was performed using a column (Synergi 4 μm Hydro-RP 80A 150 × 3.00 mm, Phenomenex, Torrance, CA, USA).

### 2.2. Compound Preparation

Cysteine trapping compound data (475 compounds) were used; 436 in-house compounds and 39 available marketed drugs were used. The composition of the experimental results showed 248 positives and 227 negatives, all of the result of (RI) integrated area is shown in Appendix A. For in-house compounds, we used compound structures registered in the Teijin database. Structural information for known compounds was obtained as 2D SDF files using ChEMBL33 [39]. It can be seen from Appendix A that the compounds used in this study mostly meet with the rule of five criteria for molecular weight (MW) and LogP [40]; and the values of polar surface area (PSA) are mostly similar to drugs which has good absorption into human body [41]. Our compounds are seemed to be drug-like and those are suitable for the dataset of this predictive model.

### 2.3. Compound Standardization through SMILES Representation

The above structural information was loaded into Maestro (Schrödinger suite 10.3) [42] and converted to canonical simplified molecular-input line-entry system (SMILES) without considering isomerization. These SMILES were used to calculate the molecular weight, polar surface area, and logP in Insight for excel [43]. These are shown as a frequency distribution graph in a Appendix A (Appendix A).

### 2.4. Classification of Data by Principal Component Analysis (PCA) Plot

To confirm that the training and test compound structures used in the random and time-split analyses were sampled from the same compound space, PCA was performed using DataWarrior (Version 5.5.0) ([44]). In DataWarrior, the compound structure in canonical SMILES format was read as input, and then FragFP was calculated as a fingerprint. Using the values of this FragFP, PCA was performed with two components.

### 2.5. Model Building with Machine Learning

#### 2.5.1. Overall Workflow

The construction of the machine learning model followed the steps depicted in Figure 1.

#### 2.5.2. Chemical Structure Fingerprints

For machine learning modeling, chemical fingerprintss and chemical structure as graphs were used. ECFP4 (2048 bit, radius: 2) [45] was calculated using the python RD-kit (2020.09.01) [46] in Chem function and AllChem, and for chemical graphs, canonical SMILES were used. Regarding chemical fingerprints, ECFP4 is a well-known fingerprint that can represent compounds better than the other fingerprints, especially for drug candidates [47]; therefore, considering its practical usage in the drug discovery process, we decided to use it.

#### 2.5.3. Objective Variable (Cysteine Trapping Posi/Nega Classification)

It has been reported that there is a correlation between the total radio isotope (RI) integrated area (unit; count) in cysteine trapping and the amount of covalent binding in microsomes (unit; pmol/g) in in vitro covalent binding tests, and there is a risk of idiosyncratic DILI at 50 pmol/g or more [18]. Considering this, the amount of covalent binding in microsomes in the in vitro covalent binding test of 50 pmol/g or more was judged as a risk. The threshold of the total RI integrated area in cysteine trapping at that time was about 1000 counts. Therefore, we labeled compounds with more than 1000 counts of RI integrated area as positive, and those with less than 1000 counts as negative.

#### 2.5.4. Machine Learning Models

We utilized the following models: Random Forests (RF) [48] using chemical fingerprints as input, and Message Passing Neural Network (MPNN) [49] using graphs as input. We used RF because it is a stable machine learning model that has already been used in various published articles [50,51,52]. MPNN was also used because excellent predictive MPNN models have been proposed in recent years [53,54,55]. The Python (version 3.7.10) scikit-learn (version 0.24.2) and the Python (version 3.7.10) Chemprop (version 1.3.1) library chemprop function were used for RF and MPNN, respectively. The parameters were set to their defaults [56].

#### 2.5.5. Model Validation

To demonstrate the validity of the model, we divided the data set using two methods for hold-out tests (random selected and time-split) and performed 5-fold cross-validation (CV). First, in random-split, 5/6 of the total data set was randomly selected as a training dataset, and the remaining 1/6 was selected as a hold-out test. Next, in time-split, the in-house compounds were sorted in the order of synthesis date, and 5/6 of the total data from the earliest was selected as a training dataset, and the remaining 1/6 of total data from the latest was selected as a hold-out test (Figure 1i–iv).

#### 2.5.6. Estimation of Important Substructures

We estimated the important substructures of the cysteine trapping assay using the MPNN model built in this study by the Monte Carlo Tree search under Chemprop framework (Figure 1v) [57]. Regarding the Monte Carlo Tree search, chemprop.interpret.py in the Python (version 3.7.10) Chemprop (version 1.3.1) was used, and the parameters were set to their defaults (Input: smiles, Number of message passing steps: 3, Batch size: 50. Number of epochs to run: 30, Learning rate: 0.001) we currently restricted the rationale to have maximum 20 atoms and minimum 8 atoms [56].

#### 2.5.7. Metrics to Compare Each In Silico Model

In the 5-fold CV, AUC-ROC and accuracy were calculated to grasp the trend of model accuracy. In the hold-out test, in addition to AUC-ROC and accuracy, the following indices (1) to (8) were calculated from the confusion matrix to examine the extrapolation of the model in detail.

TN: True Negative, FP: False Positive, FN: False Negative, TP: True Positive, MCC: Matthews Correlation Coefficient
(1)$$Accuracy=\frac{TP+TN}{TP+FP+TN+FN}$$
(2)$$Precision=\frac{TP}{TP+FP}$$
(3)$$Sensitivity=\frac{TP}{TP+FN}$$
(4)$$Specificity=\frac{TN}{TN+FP}$$
(5)Youden’s index=Sensitivity+Specificity−1
(6)$$MCC=\frac{TP\times TN-FP\times FN}{\sqrt{\left(TP+FP)(TP+FN)(TN+FP)(TN+FN\right)}}$$
(7)$$False\;rate=\frac{FN+FP}{\left(TP+FP+TN+FN\right)}$$
(8)$$F-measure=\frac{2\times Precision\times Sensitivity}{Precision+Sensitivity}$$

## 3. Results and Discussion

### 3.1. PCA Distribution of Training and Test Dataset

First, a random selection of 79 compounds was performed from the total pool of 475 compounds to assess the distribution of training and test dataset. These 79 compounds were used as an external dataset in a random-split. Additionally, the most recent 79 compounds were extracted for an external dataset as a time-split approach. PCA using FragFP was performed for the random-split and time-split to investigate the chemical space of the training and test dataset, respectively. In the random-split, the compound distribution of the training and test sets matched and was evenly distributed (Figure 2a). A bias was observed in the distribution of the test set in the time-split, likely due to recently synthesized compounds having similar targets and compound structures (Figure 2b). These findings indicate evident differences in the data selection processes between the random-split and time-split approaches, highlighting the importance of model evaluation in the time-split [58]. Furthermore, it was confirmed that the appropriate test/train selection was performed to construct a model in which the compound space remained consistent, even in the time-split, as some training compounds matched the compound distribution of the test dataset. However, it should be noted that all the compounds in the dataset are pharmaceuticals. Additionally, PCA analysis is also performed using ECFP4 as descriptor and is shown in Appendix A. (Although the conclusion was the same of FragFP, we do not discuss it due to its low contribution ratio in PC1 and PC2.)

### 3.2. Evaluation with Random-Split

To evaluate the models under the random-split, 5-fold CV was performed on both the MPNN and RF models. Table 1 shows that the ROC-AUC and accuracy of the MPNN model were 0.698 ± 0.08 and 0.668 ± 0.08, whereas those of the RF model were 0.811 ± 0.03 and 0.752 ± 0.02, respectively. Table 2 and Figure 3a,b show the prediction results for the 79 randomly extracted compounds that were included in the external dataset for each model. The ROC-AUC and accuracy of the MPNN model were 0.742 and 0.696, whereas those of the RF model were 0.819 and 0.734, respectively. Notably, the precision of the RF model was 0.789, whereas that of the MPNN model was 0.732. Both models had the same sensitivity value of 0.698. With both models surpassing an ROC-AUC of 0.70 and exhibiting favorable metrics, including sensitivity, the MPNN and RF models showed the ability to effectively predict the risk of cysteine trapping using only compound structural information. This indicates that time and cost can be reduced because experimental data are not required [49].

### 3.3. Evaluation with Time-Split

The time-split approach was used in this study because it is a more legitimate method for evaluating models than the random-split approach, which has been used in many studies recently [59,60,61]. To validate the predictive performance considering the practical aspects, a dataset of 79 recently synthesized compounds was selected as an external dataset, whereas the remaining 396 compounds were used for a 5-fold CV. The MPNN and RF models were investigated. Table 1 shows that the ROC-AUC and accuracy of the MPNN model were 0.729 ± 0.05 and 0.668 ± 0.10, while those of the RF model were 0.617 ± 0.17 and 0.556 ± 0.09 respectively, illustrating the superior accuracy of the MPNN model. The prediction results for the external dataset using the 79 chronologically extracted compounds are shown in Table 2 and Figure 3c,d. The ROC-AUC and accuracy of the MPNN model were 0.625 and 0.759, whereas those of the RF model were 0.559 and 0.671, respectively. These results indicate the higher accuracy of the MPNN model, particularly when the objective variable is imbalanced. Furthermore, the RF model exhibited a low sensitivity of 0.053, whereas the MPNN model demonstrated a higher value of 0.368, indicating that the MPNN model had an improved prediction accuracy for the compounds at a risk of generating reactive metabolites. Moreover, when evaluating the time-split dataset, the F-measure values were 0.424 and 0.071 for the MPNN and RF models, respectively, which were lower than those from the evaluation of the random-split dataset (0.714 and 0.741). However, as the F-measure is influenced by the imbalance between positive and negative objective variables in the dataset, it is assumed that this difference is due to bias of the time-split dataset (Table 3). Conversely, the MCC scores, which are independent of dataset bias, were 0.282 and −0.109 for the MPNN and RF models, respectively. The higher MCC value for the MPNN model, close to the score 0.391, indicates that the graph information captured by the MPNN model effectively extracts compound structure features without relying on dataset bias, in contrast to ECFP4. These findings align with previous research supporting the advantages of the MPNN model [53,54].

### 3.4. Estimation of Important Substructures

We determined that the MPNN model exhibited superior prediction accuracy and used it to estimate the substructures contributing to positive classification. Among the 396 compounds in the random-split training dataset, 33 market compounds were included, of which 12 yielded positive results. The MPNN model classified 6 compounds as positive correctly, of which 5 were associated with substructures reported to be involved in reactive metabolite formation. The estimated substructures of 3 compounds aligned with the literature references, whereas 2 compounds had novel substructures. In Figure 4, the structures of market drugs and putative substructures in MPNN of these five compounds are shown, along with the putative structures contributing to positive classification and the substructures reported in the literature for reactive metabolite formation.

#### 3.4.1. Propranolol

Propranolol is metabolized by CYP2D in rat liver microsomes and leads to the formation of 4-hydroxypropranolol (4-OH-PL), which is further metabolized to 1,4-naphthoquinone (1,4-NQ) [62]. The ethoxybenzene and 4-OH-PL structures inferred in this study overlapped with the structures reported in the literature.

#### 3.4.2. Verapamil

A putative substructure containing a cyano group, 2-phenylacetonitrile, was proposed for verapamil. Cyano groups generate adducts non-enzymatically, independent of the attached structure [63]. Thus, the cyano group in the estimated substructure aligns with the reported substructure.

#### 3.4.3. Imipramine

Imipramine is metabolized by CYP2D in rat liver microsomes to form 2-hydroxy imipramine, which undergoes an arene-oxide intermediate formation, and covalently binds to the CYP2D protein as a reactive metabolite [64,65]. The MPNN model classifies imipramine as positive, with a putative substructure of 5-methyl-10,11-dihydro-5H-dibenzo[b,f]azepine. This substructure inferred in this study overlap with the structure reported in the literature.

#### 3.4.4. Rosiglitazone

Rosiglitazone belongs to the thiazolidinedione class of antidiabetic drugs, along with troglitazone and pioglitazone. Troglitazone, the first therapeutic drug for insulin-resistant diabetes, was discontinued due to an idiosyncratic drug toxicity (IDT) associated with reactive metabolites. Several substructures of troglitazone, such as isocyanate-type metabolites, trigger IDT [66]. Initially, we hypothesized that the thiazolidinedione ring, which is a common structure among thiazolidinedione antidiabetic drugs, contributed to the reactive metabolites of rosiglitazone [67,68,69]. However, the MPNN model deduced a unique structure of N-ethylpyridin-2-amine, which differs from the thiazolidinedione ring. Although these three compounds exhibit covalent binding amounts of pmol/g in covalent binding assay reports [14], troglitazone and pioglitazone tested negative in the cysteine trapping assay due to the threshold area being less than 1000. In addition to the previously reported thiazolidione ring, the newly estimated N-ethylpyridin-2-amine may also contribute to the production of reactive metabolites.

#### 3.4.5. Ibrutinib

Ibrutinib, a first-in-class inhibitor of Bruton’s tyrosine kinase (BTK), irreversibly binds to cysteine 481 in the active site of BTK. 1-(Piperidin-1-yl) prop-2-en-1-one is the covalent binding site for BTK. However, the deduced structure in this study, the pyrazolopyrimidine ring, is in a different position. This difference in estimated structure arises from the fact that this model only predicts cysteine trapping scores and estimates the contributing structures [70]. As the system involves covalent bonding to BTK and differs from the experimental system of the cysteine trapping assay in this study, the pyrazolopyrimidine ring suggested here may contribute to the formation of new reactive metabolites.

## 4. Conclusions

The success rate of new drug development remains low, and DILI is often cited as a significant contributor to clinical trial failures, particularly the challenging task of predicting idiosyncratic DILI risks in the early stages of drug discovery. In this study, we focused on the cysteine trapping assay as a method for detecting reactive metabolites and developed a in silico binary classification model using in-house cysteine trapping data. To evaluate the model’s accuracy, we employed two different dataset splitting approaches: a random-split for prediction accuracy using the MPNN prediction model and the RF model, and a more practical time-split. The MPNN model exhibited strong prediction accuracy in both the random-split and time-split scenarios, indicating its ability to extract graph-based structural information even when faced with dataset biases commonly encountered in real-world drug discovery settings. Furthermore, concerning the identified substructures used for the classification of known compounds, we discovered consistent findings with substructures reported in the literature known to carry a risk of generating reactive metabolites. However, the size of dataset used in this study is relatively small for machine learning in drug discovery, raising concerns about the model’s generalizability and robustness. Expanding the dataset with more diverse chemical structures could improve the predictive performance and reliability. Nevertheless, this can contribute to the drug discovery process in a fit-for-purpose situations; for example, in the very early stage, thousands of compounds as the idea of medicinal chemists should be evaluated by this model instead of real experiments. Furthermore, this novel in silico model enables the extraction of the covalent binding risk, which is one of the factors for idiosyncratic DILI solely derived from structural information, offering potential applications in the early stages of drug discovery without the need for experimental data acquisition.

## Figures and Tables

**Figure 1 biomolecules-14-00535-f001:**
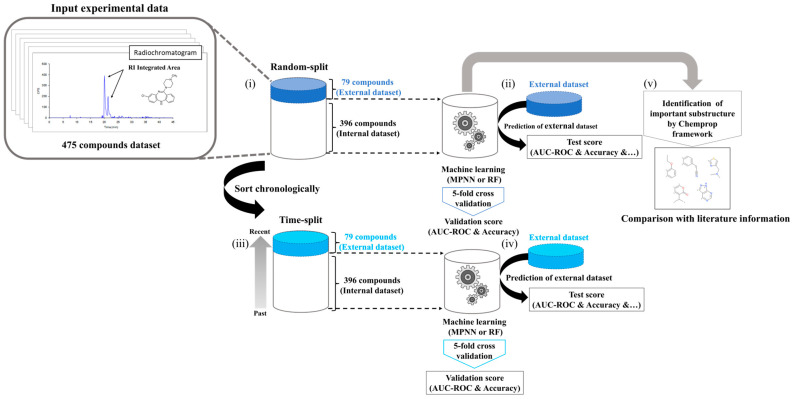
Workflow for building the cysteine trapping QSAR prediction model. (**i**) Structural information and RI (Radio Isotope) integrated area, which is counted as the radiochromatographic peaks of 475 compounds, including 436 in-house and 39 known compounds, were used for the dataset. Of the 475 compounds, 1/6 or 79 compounds were randomly extracted and used as an external dataset, and the remaining 396 compounds were used as a training dataset. (**ii**) Next, machine learning was performed on 396 compounds, 5-fold cross-validation (CV) was performed to calculate the validation score of the prediction model, and a hold-out test was performed on the external dataset to calculate the test score. (**iii**) The 475 compounds were rearranged in chronological order, and the most recent 79 compounds were extracted and used as an external dataset. (**iv**) Similar to (**iii**), machine learning was performed with 396 compounds on the time-split dataset, 5-fold CV was performed to calculate the validation score of the prediction model, hold-out test was performed on the external dataset, and the test score was calculated. (**v**) Using the dataset of 396 compounds from random splitting, we estimated the substructure used for positive determination.

**Figure 2 biomolecules-14-00535-f002:**
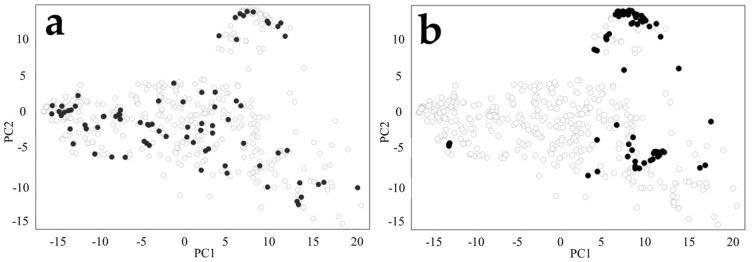
PCA plot of training and test dataset (**a**) Random-split (**b**) Time-split. White and black circles indicate compounds used in the training and external datasets, respectively. The contribution ratio of PC1 and PC2 is 26%.

**Figure 3 biomolecules-14-00535-f003:**
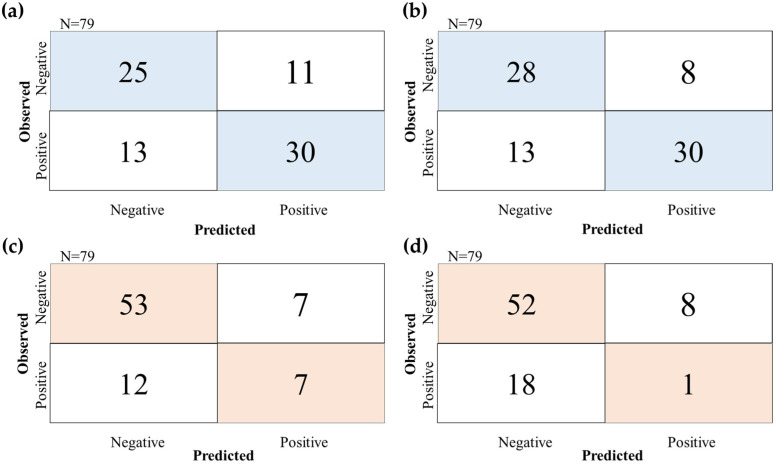
Confusion matrix of prediction with external dataset in random-split and time-split. (**a**) Prediction result of hold-out test by MPNN model at random-split, (**b**) Prediction result of hold-out test by RF model at random-split, (**c**) Prediction result of hold-out test by MPNN model at time-split, (**d**) Prediction result of hold-out test by RF model at time-split.

**Figure 4 biomolecules-14-00535-f004:**
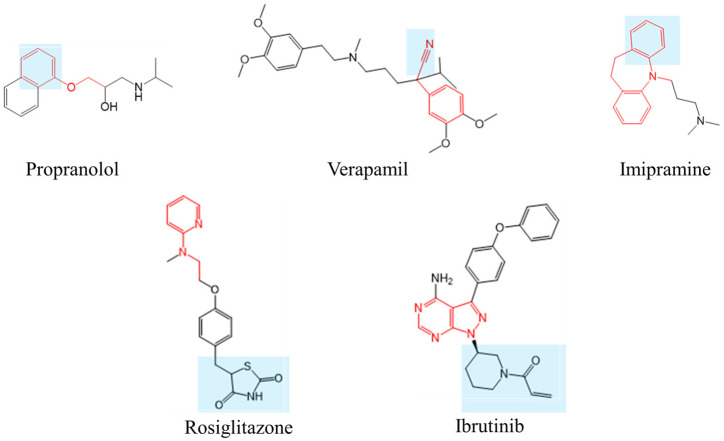
Structures of market drugs and putative substructures in MPNN. The blue highlighted part indicates the substructure reported in the literature to be involved in the generation of reactive metabolites, and the red sticks indicate the substructure predicted to have a positive contribution in the MPNN model.

**Table 1 biomolecules-14-00535-t001:** Metrics calculation for 5-fold cross validation in Message Passing Neural Network (MPNN) and Random Forest (RF) models.

Metrics	Metrics	Models					
		MPNN			RF		
Random- split	ROC-AUC	0.698	±	0.08	0.811	±	0.03
	Accuracy	0.668	±	0.08	0.752	±	0.02
Time- Split	ROC-AUC	0.729	±	0.05	0.617	±	0.17
	Accuracy	0.668	±	0.10	0.556	±	0.09
					Mean value	±	Standard deviation

**Table 2 biomolecules-14-00535-t002:** Metrics calculation for predicting external dataset of Message Passing Neural Network (MPNN) and Random Forest (RF) Models.

Dataset	Models	ROC-AUC	Accuracy	Precision	Sensitivity	Specificity	Youden’s Index	MCC	False Rate	F-Measure
Random- split	MPNN	0.742	0.696	0.732	0.698	0.694	0.392	0.391	0.304	0.714
	RF	0.819	0.734	0.789	0.698	0.778	0.475	0.474	0.266	0.741
Time- split	MPNN	0.625	0.759	0.500	0.368	0.883	0.252	0.282	0.241	0.424
	RF	0.559	0.671	0.111	0.053	0.867	−0.081	−0.109	0.329	0.071

**Table 3 biomolecules-14-00535-t003:** Composition in each positive and negative dataset.

Dataset		Positive		Negative	
Random	Train	205	(52%)	191	(48%)
	Test	43	(54%)	36	(46%)
Time	Train	229	(58%)	167	(42%)
	Test	19	(24%)	60	(76%)

## Data Availability

We can share publicly available compounds’ dataset.

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
