# Peer review of "Development of a Novel In Silico Classification Model to Assess Reactive Metabolite Formation in the Cysteine Trapping Assay and Investigation of Important Substructures"

_biomolecules, 2024, doi:10.3390/biom14050535_

Round 1

Reviewer 1 Report

Comments and Suggestions for Authors

The Article is well-designed and provides a clear presentation of results. Even though the results are not breakthroughs in the field of predicting DILI it is for sure a good tool for initial screening as the Authors suggest. They explain in a simple manner the difference in the application of RF and MPNN.
Even though the article is well written all figures are of horrible quality. All of them are of low resolution. Figure 2 requires larger x and y-axis ticks and labels (I could only read/guess) it is PC1 and PC2 after huge zoom. The same applies to the resolution of Figure S1.
The article can be accepted after the improvement of Figure quality.

Author Response

Dear Editor:

Thank you for forwarding the referees’ comments and the overall positive assessment of the work. We wish to submit a revised version of the original research article entitled: “Development of a novel in silico classification model to assess reactive metabolite formation in the cysteine trapping assay and investigation of important substructures.

Herein, we provide a rebuttal letter with point-by-point replies to #1 referee’s comments. To distinguish reviewer comments from our replies the former will be in standard font and the latter will follow a “Response (in blue)”.

We would like to thank you for the opportunity to revise our manuscript, and very much hope that the new version together with our point-by-point responses to the reviewers’ comments will be received positively by Biomolecules. If you have any further queries, please contact me.

Thank you again for considering our submission.

Yours sincerely,

Dr. Koichi Handa

DMPK Research Department

Teijin Institute for Bio-medical Research

TEIJIN PHARMA LIMITED,

4-3-2 Asahigaoka, Hino-shi,

Tokyo 191-8512, Japan

Phone: +81-42-586-8279

Fax: +81-42-587-5518

For and on behalf of the authors

Reviewer1

Comments and Suggestions for Authors

The Article is well-designed and provides a clear presentation of results. Even though the results are not breakthroughs in the field of predicting DILI it is for sure a good tool for initial screening as the Authors suggest. They explain in a simple manner the difference in the application of RF and MPNN.

Even though the article is well written all figures are of horrible quality. All of them are of low resolution. Figure 2 requires larger x and y-axis ticks and labels (I could only read/guess) it is PC1 and PC2 after huge zoom. The same applies to the resolution of Figure S1.

The article can be accepted after the improvement of Figure quality.

Response:

Thank you for your comments. We improved the Figure 2 and Figure S1. Please see the manuscript and supporting information.

Reviewer 2 Report

Comments and Suggestions for Authors

The manuscript ID biomolecules-2958830 titled “Development of a novel in silico classification model to assess reactive metabolite formation in the cysteine trapping assay and investigation of important substructures” is submitted to the Biomolecules by Umemori et al is quite an interesting study.  Predicting whether a compound can cause drug-induced liver injury (DILI) is difficult due to the complexity of the drug mechanism. The cysteine trapping assay is a method for detecting reactive metabolites that bind to microsomes covalently. However, it is cumbersome to use 35S isotope-labeled cysteine for this assay. Therefore, we constructed an in-silico classification model for predicting a positive/negative outcome in the cysteine trapping assay. We collected 475 compounds (436 in-house compounds and 39 publicly available drugs). Using a Message Passing Neural Network (MPNN) and Random Forest (RF) with extended connectivity fingerprint (ECFP) 4, we built machine learning models to predict the covalent binding risk of compounds. In the random-split dataset, the AUC-ROC values of MPNN and RF were 0.742 and 0.819 in the hold-out test, respectively. In the time-split dataset, the AUC-ROC of MPNN and RF were 0.625 and 0.559 in the hold-out test, restrictively. This result suggests that the MPNN model has a higher predictivity than RF in the time-split dataset. Hence, we conclude that the in silico MPNN classification model for the cysteine trapping assay has a better predictive power. Furthermore, most of the substructures that contributed positively to the cysteine trapping assay were consistent with previous results.

I appreciate the authors' effort to perform this study. However, the following major criticisms should be addressed before submitting the revision:

1.      The abstract is a bit confusing; Did the authors perform experimental assays to generate datasets? If so, what are the results of those assays?

2.      In the section "2.1. Experimental Method," were any experiments performed? If so, include each assay separately, such as sample preparation, enzyme assays, and analytical methods like HPLC.

3.      What is the rationale for selecting 436 in-house compounds and 39 available marketed drugs? What is the class or nature of the selected chemicals?

4.      In section "2.3. Compound Standardization," what does it mean? It's better to use the correct technical term.

5.      The authors mentioned a classification model, but there is no information about positive and negative data in the selected 475 compounds.

6.      The authors mentioned, "To confirm that the training and test compound structures used in the random and time-split analyses were sampled from the same compound space." What is the ratio of splitting?

7.      Please explain in detail how PCA was performed using DataWarrior (Ver.5.5.0).

8.      Why did the authors select only the ECFP4 fingerprint? It would be better to use 3 or more fingerprints for better prediction.

9.      Why did the authors select only RF and MPNN? Did the authors test with different ML algorithms and choose these two?

10.  Please explain in detail the Estimation of Important Structures, including input, parameters used, etc.

The current version required critical revision.

Comments on the Quality of English Language

Minor editing of English language required

Author Response

Dear Editor:

Thank you for forwarding the referees’ comments and the overall positive assessment of the work. We wish to submit a revised version of the original research article entitled: “Development of a novel in silico classification model to assess reactive metabolite formation in the cysteine trapping assay and investigation of important substructures.

Herein, we provide a rebuttal letter with point-by-point replies to #2 referee’s comments. To distinguish reviewer comments from our replies the former will be in standard font and the latter will follow a “Response (in blue)”.

We would like to thank you for the opportunity to revise our manuscript, and very much hope that the new version together with our point-by-point responses to the reviewers’ comments will be received positively by Biomolecules. If you have any further queries, please contact me.

Thank you again for considering our submission.

Yours sincerely,

Dr. Koichi Handa

DMPK Research Department

Teijin Institute for Bio-medical Research

TEIJIN PHARMA LIMITED,

4-3-2 Asahigaoka, Hino-shi,

Tokyo 191-8512, Japan

Phone: +81-42-586-8279

Fax: +81-42-587-5518

For and on behalf of the authors

Reviewer2

I appreciate the authors' effort to perform this study. However, the following major criticisms should be addressed before submitting the revision:

  1. The abstract is a bit confusing; Did the authors perform experimental assays to generate datasets? If so, what are the results of those assays?

Response:

Thank you for your comments. We performed all of the experiments as described in the manuscript. Since we understand the abstract should be described clearly, we edited the abstract and methods to include the result of positive and negative of the compounds obtained and used in this study highlighted in yellow (Line 14 to 15, and 141 to 142).

  1. In the section "2.1. Experimental Method," were any experiments performed? If so, include each assay separately, such as sample preparation, enzyme assays, and analytical methods like HPLC.

Response:

According to your comment, we made separated sections of 2.1 Experimental Methods as 2.1.1. Sample Preparation, 2.1.2. Reaction in Microsome, and 2.1.3. Measurement by HPLC with Radiomatic Detector. Especially for HPLC conditions, we described it in details. These were highlighted in yellow (Line 105, 115, 131, and 136 to 138).

  1. What is the rationale for selecting 436 in-house compounds and 39 available marketed drugs? What is the class or nature of the selected chemicals?

Response:

To be honest, we do not have an exact rationale for this selection, since we used all of the compounds whose experimental data we had. However, it can be seen from Figure S1 that the compounds used in this study meet with the rule of five criteria for MW and LogP [A]; and the values of PSA are mostly similar to drugs which have good bioavailability [B]. To sum up, our compounds are seemed to be drug-like and we think those are suitable for the dataset of the predictive model. These were described in 2.2. Compound Preparation highlighted in yellow (Line 144 to 148).

[A] Lipinski C, A. Lead- and drug-like compounds: the rule-of-five revolution. Drug Discov Today Technol. 2004, 1, 337–341. DOI: 10.1016/j.ddtec.2004.11.007, PMID: 24981612.

[B] Kelder J.; Grootenhuis P.D.; Bayada D.M.;, Delbressine L.P.;, Ploemen J.P. Polar molecular surface as a dominating determinant for oral absorption and brain penetration of drugs. Pharm Res. 1999, 16, 1514–1519. DOI: 10.1023/a:1015040217741, PMID: 10554091.

  1. In section "2.3. Compound Standardization," what does it mean? It's better to use the correct technical term.

Response:

Although compounds standardization is a correct technical term in the computer aided drug design area, to have readers understood smoothly, we edited it as “Compound Standardization through SMILES representation” highlighted in yellow (Line 149).

  1. The authors mentioned a classification model, but there is no information about positive and negative data in the selected 475 compounds.

Response:

This is a related comment to No.1 and we described the exact number of positive and negative in the abstract and method highlighted in yellow (Line 14 to 15, and 141 to 142).

  1. The authors mentioned, "To confirm that the training and test compound structures used in the random and time-split analyses were sampled from the same compound space." What is the ratio of splitting?

Response:

The ratio of splitting for test and training is 1:5 as described in Figure 1.

  1. Please explain in detail how PCA was performed using DataWarrior (Ver.5.5.0).

Response:

Although we had described how PCA was performed in DataWarrior, we edited it to have readers understood smoothly. Specifically, we added the flow of the exact action for PCA. These descriptions were highlighted in yellow (Line 158 to 160).

  1. Why did the authors select only the ECFP4 fingerprint? It would be better to use 3 or more fingerprints for better prediction.

Response:

We agree with your thoughts; we never think ECFP4 is always the best fingerprints. However, as described in the manuscript (Line 185 to 188), we selected ECFP4 since it is well-known that can represent compounds better than the other fingerprints, especially for drug candidates [C]. The reason why we did not used other fingerprints is that we would like to focus on the practical usage of the predictive models; in this sense ECFP4 is commonly used in our daily research life and ideal one to be investigated.

[C] O’Boyle, N.M.; Sayle, R.A. Comparing structural fingerprints using a literature-based similarity benchmark. J. Cheminform. 2016, 8, 36. DOI:10.1186/s13321-016-0148-0, PMID: 27382417.

  1. Why did the authors select only RF and MPNN? Did the authors test with different ML algorithms and choose these two?

Response:

We did not investigate the other models. As described in the manuscript (Line 201 to 203), we think RF is one of the reliable ML models like a golden standard and MPNN is one of the cutting edge algorithms using graphs. We know that there are many reports that used and investigated many algorithms; however, we would like to investigate and focus on the usage of practical ways in the drug discovery stages rather than theoretical investigation.

  1. Please explain in detail the Estimation of Important Structures, including input, parameters used, etc.

Response:

According to your comment, we added the detailed settings in 2.5.6. Estimation of Important Substructures highlighted in yellow (Line 220 to 222).

Comments on the Quality of English Language Minor editing of English language required.

Response:

This manuscript has been proofread by the English proofreading company Editage (https://www.editage.com/).

Round 2

Reviewer 2 Report

Comments and Suggestions for Authors

I appreciate the author for organizing each experimental section separately. However, the results of those analyses are missing. For example, the results of "Reaction in Microsome" and "Measurement by HPLC with Radiomatic Detector" are not provided.

The authors mentioned that, "To be honest, we do not have an exact rationale for this selection, since we used all of the compounds whose experimental data we had." Consequently, it becomes challenging to evaluate the study. I suggest the authors deposit their compounds in relevant databases such as PubChem, ZINC, or DrugBank and utilize authentic compound IDs. If the compound IDs exist, they should be included as an appendix. Generally, open-access publishers provide all resources to ensure reproducibility without intellectual property issues.

How did you define positive and negative data? Which values were used? This should be clearly stated.

The statement "FragFP was calculated as a fingerprint" needs clarification. What exactly is FragFP? Additionally, if the authors used different fingerprints for PCA and ML models, the reasoning behind this decision should be explained.

Author Response

Dear Editor:

Thank you for forwarding the referees’ comments and the overall positive assessment of the work. We wish to submit a revised version of the original research article entitled: “Development of a novel in silico classification model to assess reactive metabolite formation in the cysteine trapping assay and investigation of important substructures.

Herein, we provide a rebuttal letter with point-by-point replies to #2 referee’s comments. To distinguish reviewer comments from our replies the former will be in standard font and the latter will follow a “Response (in blue)”.

We would like to thank you for the opportunity to revise our manuscript, and very much hope that the new version together with our point-by-point responses to the reviewers’ comments will be received positively by Biomolecules. If you have any further queries, please contact me.

Thank you again for considering our submission.

Yours sincerely,

Dr. Koichi Handa

DMPK Research Department

Teijin Institute for Bio-medical Research

TEIJIN PHARMA LIMITED,

4-3-2 Asahigaoka, Hino-shi,

Tokyo 191-8512, Japan

Phone: +81-42-586-8279

Fax: +81-42-587-5518

For and on behalf of the authors

Reviewer2

I appreciate the authors' effort to perform this study. However, the following major criticisms should be addressed before submitting the revision:

Response:

We appreciate for your comments and repeated reviewing. It is very helpful to improve our manuscript. We did our best to answer your questions and comments. We would be glad if you could embrace our responses.

I appreciate the author for organizing each experimental section separately. However, the results of those analyses are missing. For example, the results of "Reaction in Microsome" and "Measurement by HPLC with Radiomatic Detector" are not provided.

Response:

Thank you for your comments. According to your comment, we provided all of the result of the total radio isotope (RI) integrated area in the supplementary material as Table S1. This description was added in the Line 142 to 143.

The authors mentioned that, "To be honest, we do not have an exact rationale for this selection, since we used all of the compounds whose experimental data we had." Consequently, it becomes challenging to evaluate the study. I suggest the authors deposit their compounds in relevant databases such as PubChem, ZINC, or DrugBank and utilize authentic compound IDs. If the compound IDs exist, they should be included as an appendix. Generally, open-access publishers provide all resources to ensure reproducibility without intellectual property issues.

Response:

Thank you for your comments. I understand your comments; however, we utilize in-house compounds of pharmaceutical company. In this case it is impossible to open the chemical structures due to the problem of patent. On the other hand, according to your comments, we disclosed molecular properties (Figure S1) and the experimental values of RI integrated area (Table S1). Generally, there are so many articles in various journals which includes hidden chemical structures from pharmaceutical companies including our previous ones [A-C]. Those are very informative and contribute to the science from the viewpoint of in silico method development and drug discovery strategy. We think this kind of research should be allowed to be published in the journals not to avoid the progress of scientific development. Consequently, again we understand your points; however, we would be really glad you could embrace it.

[A] Handa K, Wright P, Yoshimura S, Kageyama M, Iijima T, Bender A. Prediction of Compound Plasma Concentration-Time Profiles in Mice Using Random Forest. Mol Pharm. 2023 Jun 5;20(6):3060-3072. doi: 10.1021/acs.molpharmaceut.3c00071. Epub 2023 Apr 25. PMID: 37096989; PMCID: PMC10245373.

[B] Handa K, Thomas MC, Kageyama M, Iijima T, Bender A. On the difficulty of validating molecular generative models realistically: a case study on public and proprietary data. J Cheminform. 2023 Nov 21;15(1):112. doi: 10.1186/s13321-023-00781-1. PMID: 37990215; PMCID: PMC10664602.

[C] Umemori Y, Handa K, Sakamoto S, Kageyama M, Iijima T. QSAR model to predict Kp,uu,brain with a small dataset, incorporating predicted values of related parameter. SAR QSAR Environ Res. 2022 Nov;33(11):885-897. doi: 10.1080/1062936X.2022.2149619. PMID: 36420623.

How did you define positive and negative data? Which values were used? This should be clearly stated.

Response:

We had described the criteria (1,000 counts of total RI integrated area) of positive and negative in the Line 197 to 198.

The statement "FragFP was calculated as a fingerprint" needs clarification. What exactly is FragFP? Additionally, if the authors used different fingerprints for PCA and ML models, the reasoning behind this decision should be explained.

Response:

The default descriptor in DataWarrior, FragFp, is a binary fingerprint based on a dictionary of substructure fragments, akin to the MDL keys. It operates using a set of 512 predefined structural fragments. This fingerprint has been already used in many articles [D-F]. Then, according to your comment of ECFP, we calculated another PCA using ECFP4 and this was attached in the supplementary material as Table S2. However, since the cumulative contribution ratio was very low, we could not lead the exact conclusion from it. This description was added in the Line 257 to 259.

[D] von Korff M, Sander T. Limits of Prediction for Machine Learning in Drug Discovery. Front Pharmacol. 2022 Mar 10;13:832120. doi: 10.3389/fphar.2022.832120. PMID: 35359835; PMCID: PMC8960959.

[E] Ertl P, Patiny L, Sander T, Rufener C, Zasso M. Wikipedia Chemical Structure Explorer: substructure and similarity searching of molecules from Wikipedia. J Cheminform. 2015 Mar 22;7:10. doi: 10.1186/s13321-015-0061-y. PMID: 25815062; PMCID: PMC4374119.

[F] Anastasia A Nikitina, Alexey A Orlov, Liubov I Kozlovskaya, Vladimir A Palyulin, Dmitry I Osolodkin, Enhanced taxonomy annotation of antiviral activity data from ChEMBL, Database, Volume 2019, 2019, bay139, https://doi.org/10.1093/database/bay139
